# An Improved Detection Algorithm for Ischemic Stroke NCCT Based on YOLOv5

**DOI:** 10.3390/diagnostics12112591

**Published:** 2022-10-26

**Authors:** Lifeng Zhang, Hongyan Cui, Anming Hu, Jiadong Li, Yidi Tang, Roy Elmer Welsch

**Affiliations:** 1School of Information and Communication Engineering, Beijing University of Posts and Telecommunications, Beijing 100876, China; 2State Key Laboratory of Networking & Switching Technology, Beijing University of the Posts and Telecommunications, Beijing 100876, China; 3Department of Rehabilitation Medicine, Beijing Tiantan Hospital, Capital Medical University, Beijing 100070, China; 4Sloan School of Management, Massachusetts Institute of Technology, Cambridge, MA 02139, USA; 5Center for Statistics and Data Science, Massachusetts Institute of Technology, Cambridge, MA 02139, USA

**Keywords:** ischemic stroke, noncontrast computer tomography, features, detection algorithm, YOLOv5

## Abstract

Cerebral stroke (CS) is a heterogeneous syndrome caused by multiple disease mechanisms. Ischemic stroke (IS) is a subtype of CS that causes a disruption of cerebral blood flow with subsequent tissue damage. Noncontrast computer tomography (NCCT) is one of the most important IS detection methods. It is difficult to select the features of IS CT within computational image analysis. In this paper, we propose AC-YOLOv5, which is an improved detection algorithm for IS. The algorithm amplifies the features of IS via an NCCT image based on adaptive local region contrast enhancement, which then detects the region of interest via YOLOv5, which is one of the best detection algorithms at present. The proposed algorithm was tested on two datasets, and seven control group experiments were added, including popular detection algorithms at present and other detection algorithms based on image enhancement. The experimental results show that the proposed algorithm has a high accuracy (94.1% and 91.7%) and recall (85.3% and 88.6%) rate; the recall result is especially notable. This proves the excellent performance of the accuracy, robustness, and generalizability of the algorithm.

## 1. Introduction

Cerebral Stroke (CS) is a heterogeneous syndrome caused by multiple disease mechanisms and causes a disruption of cerebral blood flow with subsequent tissue damage [1,2]. CS is an acute cerebrovascular disease with a high incidence rate, high mortality rate, and high disability rate [3]. CS occurs when the blood supply in the brain is damaged. It is a result that blocks cerebral vessels and stops blood from flowing to an area of the brain (called an ischemic stroke) or bursts cerebral vessels and causes intracerebral hemorrhage (called a hemorrhagic stroke). Every year, 15 million people suffer from stroke all over the world, including 5 million deaths and 5 million permanent disabilities. Survivors can experience loss of vision and/or speech, paralysis, and confusion [4]. Investigations have shown that CS is the primary reason for adult disability in some countries [2].

The incidence rate of ischemic stroke (IS) is higher at 60~70% than CS. The most common symptoms of IS are a sudden weakness of the face, arms, or legs, sudden dizziness and unconsciousness, sudden mouth/eye deviation, hemiplegia, confusion, difficulty in speaking or understanding, monocular or binocular vision difficulty, difficulty in walking, dizziness, loss of balance or coordination, severe headache without cause, fainting, etc. [5,6,7]. The severity and symptom duration of neurological dysfunction after cerebral artery stenosis and occlusion can be divided into three types: transient ischemic attack, reversible ischemic neurologic deficit, and complete stroke, among which the symptoms of a complete stroke are the most serious [8]. After the onset of IS, it is painful for patients, their families, and society. IS is a cerebrovascular disease with a high incidence in middle age, mostly in people over 40 years old.

At present, prevention is considered to be the best measure for IS because of the lack of effective treatment. For the prevention of stroke, effective prevention strategies include reducing the intake of salt in a diet, eating more fresh vegetables and fruits, strengthening an appropriate amount of physical exercise, avoiding excessive drinking, and banning smoking, etc., in daily life, which has been proven to effectively reduce the incidence of cardiovascular disease. In addition, the regular screening of healthy people and regular monitoring and intervention of high-risk populations with disease risk have also proved to be effective. Necessary intervention measures are proposed for potential patients too. Timely and accurate diagnosis plays an important role in preventing and developing a treatment plan. The clinical diagnosis includes detection in blood and urine, electrocardiogram and blood pressure, cerebral angiography, and brain imaging technology, in which brain imaging technology plays an important role in the cause, location, and area of a stroke [9]. Brain imaging technology for imaging brain tissue or structures utilizes rays (X-ray, γ, etc.) or radio waves, such as noncontrast computer tomography (NCCT), CT angiography (CTA), CT perfusion (CTP), magnetic resonance imaging (MRI). The imaging duration of NCCT is less than one hour at the fastest, and the cost is half or even lower than other brain imaging technologies [10]. Therefore, NCCT is actually a common detection method for stroke, especially for acute IS [11,12].

Computer-aided diagnosis (CAD) is increasingly becoming a reliable and essential means against the background of artificial intelligence (AI) [13,14,15,16,17,18]. For the region of interest (ROI) detection, the current CAD models detect the ROI by mining valuable features in the images. Consequently, the more obvious the features of the ROI, the better the detection. However, the characteristics of NCCT for acute IS are relatively simple, whether it is the composition elements or the color in the image area, especially in the ROI, the hypodensity in the infarct area, the gray matter, types of gray matters, etc. [19], which have differences compared to some diseases, such as tumors with obvious tissue deformation; this leads to a lower feature detection sensitivity of NCCT for IS [20]. Chalela et al. evaluated NCCT to screen acute ischemic stroke. The result was only 56 acute IS patients were screened from 217 acute IS patients, with a sensitivity of 26% [21]. Lansberg et al. also found that the screen rate of acute IS from NCCT was 42–63% [22]. Marbun et al. proposed a method that combined contrast-limited adaptive histogram equalization (CLAHE) and CNN to detect stroke NCCTs and achieved 90% accuracy and 76% sensitivity using a small test set [23]. Kuang et al. studied a model based on a random forest classifier to detect 100 testing acute IS patient NCCTs with a specificity of 91.8% and a sensitivity of 66.2% [24]. Luo et al. proposed the UCATR for the detection of acute IS lesions. UCATR uses a transformer to learn the global features of NCCT, and CNN is used to detect the ROI. The result showed good performance, with a Dice similarity coefficient of 73.58% and a sensitivity of 73.12% [25]. Therefore, we studied a detection algorithm that can realize high accuracy and sensitivity detection for IS NCCT.

## 2. Materials and Methods

### 2.1. Image Enhancement

The enhanced image is a kind of data transformation for image data in order to highlight some feature information, creating a better visual effect that can easily fit the subsequent image processing. Its purpose is to make some features that are not obvious in the original image easier to interpret and recognize. These features are often selective. At the same time, the way to enhance them is relative, which can enhance the feature information of interest or weaken the feature information of interest.

The common methods of enhancement can be divided into the spatial domain and frequency domain, according to the processing spaces of the target object [26,27]. The objective of the spatial method is the enhancement of the image itself. The operation is for pixels and components composed of pixels in the image, including the grayscale transformation, histogram equalization, and the smoothing of image areas; the frequency method refers to the enhancement or suppression of some features through processing for the frequency domain. For example, reducing noise by setting a low-pass filter.

As one of the most classic spatial enhancement methods, histogram equalization (HE) is an algorithm that uses the pixel conversion of an image to increase image contrast [28]. Take the grayscale image as an example; firstly, create a histogram of the grayscale values in the selected area; secondly, convert the grayscale histogram into an approximate uniform distribution and, finally, transform the grayscale histogram into an image based on the approximate uniform distribution (according to mapping function) to achieve an image contrast adjustment. With the deepening of research into different application scenarios, some new derivative algorithms of histogram equalization have been proposed, such as adaptive histogram equalization (AHE) [29], contrast-limited adaptive histogram equalization (CLAHE) [30], and adaptive contrast enhancement using the local region stretching [31]. The adaptive contrast enhancement using local region stretching algorithm (ACELES) firstly divides the image according to its brightness and then performs HE on the three regions; then, after the HE of the three regions, the equalization result and the input are weighted averages to realize the output of the result. In addition, grayscale stretching is an image enhancement method that uses grayscale transformation via a linear transformation function. Its main idea is to improve the dynamic range of the grayscale levels in image processing [32]. Zhang et al. studied an image enhancement algorithm based on K-means, which was used to deal with imaging in low-brightness environments [33].

### 2.2. The Model of Detection Based on YOLOv5

Conventional convolutional neural networks (CNN), such as region CNN (R-CNN) and fast region CNN (fast R-CNN), are two-stage target detection algorithms [34,35]. In the two-stage detection, the model will extract the features of the detection region and generate the region proposal (RP) during the first stage, and then use the CNN to classify the samples in the second stage. The YOLO (you only look once) series as deep learning with one stage has been proposed since 2016 and has developed into YOLOv5 (the fifth vision YOLO algorithm). YOLOv5 is the one of state-of-the-art algorithms in the YOLO family, which is one of the most popular target detection algorithms at present and is applied in target detection and recognition [36,37]. The YOLO algorithm is a one-stage target detection algorithm, which the model can directly use for the network structure to extract features and realize the detection and classification of samples, leaving out the process of RP. Therefore, the YOLO model is more lightweight and efficient, relying on the detection accuracy and generalization ability [38,39,40].

YOLOv5 is a kind of CNN with the inherent network composition structure of a traditional CNN, including the input layer, convolution layer, and pooling layer. In addition, it also adds new network composition modules [41,42]. The structure of YOLOv5 is shown in Figure 1. It mainly consists of four parts: Input, Backbone, Neck, and Head. The image is input into YOLOv5 through the Input Layer and is completed by the image preprocessing, including normalization, data enhancement, and the setting of the bounding box. The function of the Backbone network is to realize the learning and classification of the image features. It includes one convolution component, the focus component, with one convolution module and several residual network modules. The Neck network is used to improve the diversity of the features and the robustness of the network while acquiring the features of different stages. The design of the head output is used to realize the output of the target detection results. The structure of YOLOv5 is shown in Figure 1.

Specifically, the image inputs the YOLOv5 network from the Input Layer. The Input Layer mainly completes the basic preprocessing operation, including image compression and normalization. The process of normalization is completed with the Batchnorm2d module, as shown in Equation (1); in addition, YOLOv5 uses the Mosaic data enhancement operation in the Input Layer to improve the training speed of the model and the accuracy of the network. In addition, the YOLOv5 model also adds an adaptive anchor frame module and an adaptive picture scaling module. The function of the Backbone network is to realize the learning and classification of the image features. It includes a csparknet53 structure and a focused structure, including one convolution module and 23 residual network modules; the Neck network can improve the variety of features and the robustness of the network by acquiring features at different stages. It includes an important feature fusion module: FPN-PAN. The FPN structure includes three maximum pooling layers, which can enhance the fusion of different features. The PAN can further enhance the fusion of the FPN features. In the FPN-PAN structure, the deep semantic features can be transmitted to the shallow layer through FPN, and the shallow positioning information can be transmitted to the deep layer through the PAN, thus enhancing the multi-scale and multi-level feature map fusion. In the process of learning and acquiring features, the Backbone and Neck networks need to activate functions to fit the features. The activate function is called Leaky_ReLU, as shown in Equation (2). The Head output is located in the last layer of the network. It is the output of the target detection result. It includes three detections using the convolution structure and detects the target on the feature map from the previous network, finally outputting the detection result. At this time, the Sigmoid as an activation function will also be used, as shown in Equation (3).
(1)$$n=\frac{m-E\left[m\right]}{\sqrt{Var\left[m\right]+ο}}•\alpha +\beta$$
where, *m* is the original data to be normalized, $E\left[m\right]$ and $Var\left[m\right]$, respectively, represent the mean and variance of *m*, *n* is the normalized data, α and β are the parameters for linear transformation; the parameter ο can avoid meaningless equations when the $Var\left[m\right]$ is equal to zero.
(2)$$y_{i}=\left\{\begin{array}{c} \frac{x_{i}}{a_{i}},x_{i}<0\\ x_{i},x_{i}\geq 0 \end{array}\right.$$
where *i* represents the coefficient of the *i*th hidden layer, and *x_i_* and *y_i_*, respectively, represent the input and output of the *i*th hidden layer, after being processed by the activation function $a_{i}\in \left(1,+\left.\infty \right)\right.$.
(3)$$f\left(x\right)=\frac{1}{1+e^{-x}}$$
where *x* and *y*, respectively, represent the input and output in output layer.

### 2.3. AC-YOLOv5

In this paper, we propose an improved detection algorithm for IS NCCT based on YOLOv5 (AC-YOLOv5). The original IS NCCT is enhanced by ACELES; then, the enhanced image input to the model based on YOLOv5 is for target detection. The Algorithm 1 is as follows.

**Algorithm 1.** AC-YOLOv5Input: The original image;Output: The Network structure, location, and accuracy of the target(s);Step 1: Adaptively acquire the gray values of pixel points in the original image, and divide all pixel points into three classes based on the gray values, i.e., light, medium and dark according to the gray values;Step 2: Perform HE on the three classes respectively, and then weight the results after HE and output the image;Step 3: Input the image into the YOLOv5. Generate the original feature maps and feature matrix *A* about the object(s) based on the image;Step 4: Training to predict and get feature maps and feature matrices $\hat{A}$;Step 5: The loss function is calculated and optimized according to the matrix A and the matrix $\hat{A}$;Step 6: Perform iterations from step 3 to step 5 until the desired accuracy is achieved.

The structure of AC-YOLOv5 is shown in Figure 2. Firstly, the model performs ACELES to achieve image enhancement. This step leads to the features of ROI of NCCT becoming more prominent in the global region. The NCCT image is a grayscale image; thus, the grayscale value of a pixel actually represents the brightness of the pixel; Secondly, Input the enhanced image into YOLOv5. The image is divided into several grids of equal size and generates the bounding box(es) of the object(s). The original feature maps for the objects are generated too, which are used to store three feature vectors for the object in the image in the grid, including object position information (center point, height, and the width of the box), the class is inside or outside the grid and the class information. Thirdly, training to obtain the anchor point and box of the object(s); then, predict and get feature maps, including the center point and height and width of the box. The loss function is calculated and optimized according to the original feature maps and the predicted feature maps; finally, repeat the second and third steps until the network convergence condition is reached. Output the position information of the target objects and the width and height of the bounding box(es).

In terms of convergence conditions, we use the loss function as the preset. The loss function consists of three parts: the bounding box deviation, the confidence deviation, and the prediction accuracy deviation, which are shown in Equation (4). There are five terms of polynomial accumulation in the formula, wherein the sum of the first term and the second item describes the position information of the bounding box, the third term and the fourth term describe the confidence information, and the fifth term describes the deviation of the prediction probability.
(4)$$\begin{array}{c} Loss=\lambda_{\mathrm{coord}}\sum_{i=0}^{s^{2}}\sum_{j=0}^{B}1_{ij}^{obj}\left[{\left(x_{i}-{\hat{x}}_{i}\right)}^{2}+{\left(y_{i}-{\hat{y}}_{i}\right)}^{2}\right]\\ +\lambda_{\mathrm{coord}}\sum_{i=0}^{s^{2}}\sum_{j=0}^{B}1_{ij}^{obj}\left[{\left(\sqrt{w_{i}}-\sqrt{{\hat{w}}_{i}}\right)}^{2}+{\left(\sqrt{h_{i}}-\sqrt{{\hat{h}}_{i}}\right)}^{2}\right]\\ +\sum_{i=0}^{s^{2}}\sum_{j=0}^{B}1_{ij}^{obj}{\left(C_{i}-{\hat{C}}_{i}\right)}^{2}\\ +\lambda_{noobj}\sum_{i=0}^{s^{2}}\sum_{j=0}^{B}1_{ij}^{noobj}{\left(C_{i}-{\hat{C}}_{i}\right)}^{2}\\ +\sum_{i=0}^{S^{2}}1_{i}^{obj}\sum_{c\in classes}{\left(p_{i}\left(c\right)-{\hat{p}}_{i}\left(c\right)\right)}^{2} \end{array}$$
where *x_i_*, *y_i_*, respectively, represent the abscissa and ordinate of the center point of the anchor box, *h_i_* is the width of the anchor box, *w_i_* is the height of the anchor box, *C_i_* is the confidence score for the object inside or outside the anchor box, *p_i_*(*c*) is probability of classification. $1_{ij}^{obj}$ represents the *j*th predictor in the *i*th box. If there is an object in the area and the confidence of *j*th predictor is the highest, then the $1_{ij}^{obj}$ is 1. On the contrary, if there is no object in the *i*th cell, then the $1_{ij}^{noobj}$ is 1. $\lambda_{coord}$ and λnoobj are constants, and the weight of related terms can be adjusted.

## 3. Results

### 3.1. Dataset and Construction of Detection Model

The experimental data for the IS NCCT images we selected consisted of two kinds. One is from GitHub, which is one of the most popular software open-source websites at present. We selected the Hypodense-Segmentation-Using-CNN and AISD datasets. The first dataset contains 250 CT images. The second dataset, AISD, contains 397 CT images of IS, all of which are from IS patients and have been labeled for the ROI regions. The second kind of datat came from open-source website resources [43,44,45,46,47], and the ROI region labeling was also already carried out. The multisource experimental data can better verify the performance of the model, including accuracy and generalization ability.

In terms of the construction of the detection model, the dataset of Hypodense-Segmentation-Using-CNN was selected to complete this process. The dataset is divided into three parts, including the training dataset, validation dataset, and testing dataset. The ratio was 3:1:1, that is, 150 training samples, T, 50 verification samples, V, and 50 test samples, T1. Otherwise, we selected 64 samples from AISD and open website resources to compose the test sample, T2, to verify the accuracy and generalization ability of the model.

### 3.2. Results and Evaluation

As described in Figure 3, there were training results in the YOLOv5 model’s own results window. There were some items in the results window, including three feature vectors (box (the position information of object, i.e., center point, height, and width of box), obj (inside or outside the grid), cls (class information)), metrics (precision, recall, and mAP (mean Average Precision)). After hundreds of epochs, the three feature vectors and the metrics reach the convergence conditions set by the loss function. This indicates that the AC-YOLOv5 model shows excellent performance, including the accuracy and robustness of the model.

In order to better evaluate the performance of the AC-YOLOv5 model, we added seven control group experiments: Fast R-CNN (FR-CNN), YOLOv5(without image enhancement), the image enhancement based on Kmeans + YOLOv5 (K-YOLOv5), the image enhancement of Grayscale stretching + YOLOv5 (G-YOLOv5), the image enhancement of HE + YOLOv5 (H-YOLOv5), the image enhancement of AHE + YOLOv5 (AH-YOLOv5), and the image enhancement of CLAHE + YOLOv5 (CL-YOLOv5). In addition, we used Accuracy and Recall to evaluate the accuracy of the algorithm. The generalization was verified by T1 and T2.

The experimental results for T1 and T2 are shown in Table 1. The experimental results show that image enhancement based on HE can significantly improve the accuracy of detection and the generalization ability of the model. Moreover, K-YOLOv5 had the worst results, whether that was for T1 or T2. AC-YOLOv5 had the best comprehensive performance. Specifically, the accuracy of the proposed AC-YOLOv5 algorithm on the T1 and T2 sets was 94.1% and 91.7%, and the recall rate was 85.3% and 88.6%, respectively, especially in generalization ability.

## 4. Discussion

Research shows that timely and rapid detection of IS and targeted interventions can minimize the damage and even save the patient’s life. As a fast-imaging technology, NCCT is one of the most important methods to detect IS [48]. In this paper, we presented a detection algorithm for IS NCCT. The algorithm enhances the features of NCCT to make the ROI easier to detect, and the results show high accuracy and recall performance.

Our proposed algorithm includes an image enhancement module and a detection module. The image enhancement module introduces the adaptive mechanism in the enhancement of NCCT, which makes feature enhancement processing become more adaptive and generalizable. Meanwhile, the image enhancement module can make the local features more prominent rather than global, potentially useless features, which will undoubtedly improve the subsequent detection accuracy. In the detection module, ROI detection is completed based on the YOLOv5 target detection model. YOLOv5, as one of the most advanced target detection algorithms, has a lighter network structure and stronger generalization capability than traditional CNN networks. It has been proven to be an efficient target detection model in many fields [49,50].

Our algorithm has achieved excellent accuracy and recall performance, especially for recall. Other studies have also focused on solving the problem of the insufficient sensitivity of IS NCCT features. From their research results, recall is often an urgent problem, and the verification of the model is mainly completed on a single dataset [51]. Our algorithm has achieved good results and has been verified over several datasets. We propose that AC-YOLOv5 achieves 94.1% accuracy and 85.3% recall and 91.7% accuracy and 88.6% recall on the two different datasets, respectively; it has the best comprehensive performance when compared to the seven control groups in the experiment. In addition, from the experimental results, we also found that the YOLOv5 algorithm has better overall performance than traditional CNN algorithms; the model training of CNN training needs a lot of data as a basis [12], and this will also add additional time costs. Both the traditional CNN algorithm and the YOLOv5 algorithm have the problem of insufficient sensitivity toward feature detection in NCCT, but image enhancement based on HE can effectively solve this problem.

## 5. Conclusions

CS is a serious disease that threatens human health and normal life, especially IS, which accounts for the majority of CS cases. NCCT is one of the most important methods for IS detection. The AC-YOLOv5 model can effectively solve the problem of feature insensitivity in the CAD of IS NCCT and achieve efficient diagnosis effects with accuracy and generalizability. We also hope to provide a feasible and effective exploration for the CAD of NCCT in other diseases to save time and human resources.

## Figures and Tables

**Figure 1 diagnostics-12-02591-f001:**
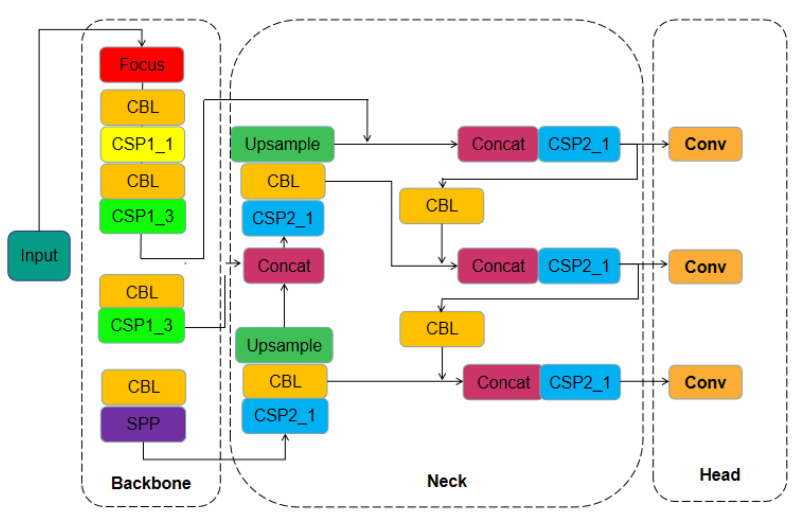
The architecture of YOLOv5.

**Figure 2 diagnostics-12-02591-f002:**
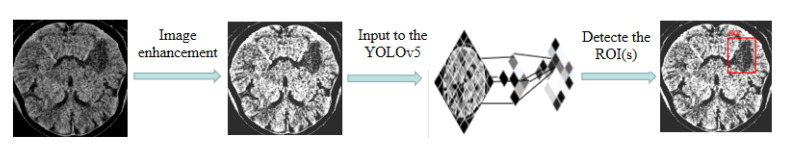
The structure of AC-YOLOv5.

**Figure 3 diagnostics-12-02591-f003:**
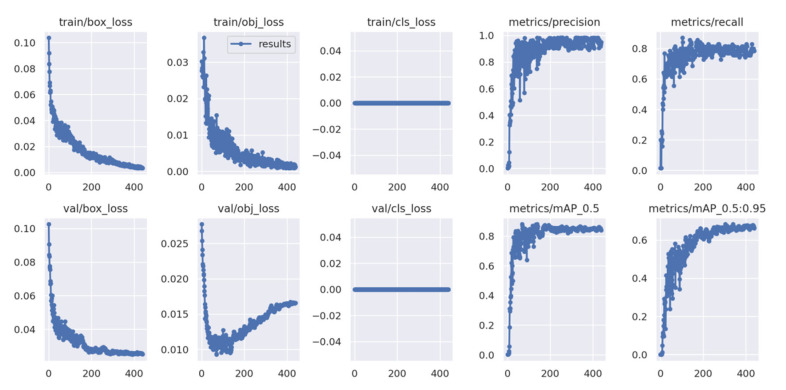
The results of AC-YOLOv5 model training.

**Table 1 diagnostics-12-02591-t001:** The experimental results on T1 and T2.

	T1		T2	
	Accuracy	Recall	Accuracy	Recall
F-RCNN	67.80%	54.30%	2.00%	4.10%
YOLOv5	83.10%	67.50%	55.10%	40.40%
K-YOLOv5	34.30%	22.00%	7.60%	7.61%
G-YOLOv5	86.30%	69.00%	57.10%	42.20%
H-YOLOv5	87.10%	65.00%	77.60%	65.10%
AH-YOLOv5	98.70%	76.10%	74.90%	69.60%
CL-YOLOv5	87.00%	71.40%	57.70%	65.20%
**AC-YOLOv5**	94.10%	85.30%	91.70%	88.60%

## Data Availability

Publicly available datasets were analyzed in this study. This data can be found here https://github.com/bhetrisonia/Hypodense-Segmentation-using-CNN; https://github.com/GriffinLiang/AISD; https://emedicine.medscape.com/article/338385-overview; https://medizzy.com/feed/14746660; https://pmj.bmj.com/content/86/1017/409.short; https://neuropedia.net/ischemic-stroke/articles/neurology/cerebrovascular/; https://www.medical.theclinics.com/article/S0025-7125(18)30123-8/fulltext (accessed on 1 September 2022).

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
