# Peer review of "An Improved Detection Algorithm for Ischemic Stroke NCCT Based on YOLOv5"

_diagnostics, 2022, doi:10.3390/diagnostics12112591_

Round 1
Reviewer 1 Report
The paper has many flaws that harden its readability
there is no background and related work section
the equations presentation is very bad
the explanation of the model is shallow
the results disscusion is not sufficient
the conclusion is very naive
Author Response
The paper has many flaws that harden its readability
Comment 1: There is no background and related work section.
Response 1: We would like to express our sincere appreciation for your careful reading and invaluable comments. We have studied the comments carefully and have made corrections which we hope to meet with approval. We added background and related work on page 2 in yellow color, and updated relevant references.
Comment 2: The equations presentation is very bad.
Response 2: Thank you for pointing it out. We revised the relevant presentation about the equations. The relevant contents are highlighted on pages 3 & 6 of the manuscript in yellow color, updated relevant references on pages 3 & 6.
Comment 3: The explanation of the model is shallow.
Response 3: Thank you very much for the valuable comment. According to the comment, we added relevant contents are highlighted on pages 4 & 5 of the manuscript in yellow color.
Comment 4: The results discussion is not sufficient.
Response 4: Thank you very much for the significant comment. According to comment, we rewrote the discussion section. The relevant contents are highlighted on page 8 of the manuscript in yellow color, and updated relevant references.
Comment 5: The conclusion is very naive.
Response 5: Thank you very much for the significant comment. We carefully studied the comment and we rewrote the conclusion section. The relevant contents are highlighted on page 8 of the manuscript in yellow color.

Reviewer 2 Report
This manuscript has provided sufficient background for CNN algorithms for NCCT for IS detection. YOLOv5 performs better than conventional CNN algorithm. Image enhancement based on HE could solve the problem of insufficient sensitivity to feature detection. The proposed algorithm has demonstrated excellent detection accuracy and generalization.
There are only a few typos to be corrected. They are listed here:
line 107: adaptive contract enhancement--> adaptive contrast enhancement
line 113: Zhang et al. Studied-->Zhang et al. studied
Author Response
This manuscript has provided sufficient background for CNN algorithms for NCCT for IS detection. YOLOv5 performs better than conventional CNN algorithm. Image enhancement based on HE could solve the problem of insufficient sensitivity to feature detection. The proposed algorithm has demonstrated excellent detection accuracy and generalization.
Comment 1: There are only a few typos to be corrected. They are listed here:
line 107: adaptive contract enhancement--> adaptive contrast enhancement
line 113: Zhang et al. Studied-->Zhang et al. studied
Response 1: Thank you for your meticulous review and comments. The relevant contents are highlighted on page 3 of the manuscript in green color.

Reviewer 3 Report
1. This study proposes an improved YOLOv5 detection algorithm that can enhance the features in the image to make them more effective in mine and detect.
2. Experimental results show that the proposed AC-YOLOv5 has achieved excellent results in accuracy, recall, and generalization of the model compared with the other 7 control groups experiment, especially the generalization for different datasets.
3. Figure 1 is unclear
4. In Figure 3, coordinate axis units are ambiguous, and the specifications should be consistent for better comparison.
5. Figure 3 needs to be detailed in the text to improve readability.
6. It is recommended to elaborate in the Discussion what factors or differences make AC-YOLOv5 more accurate than the other 7 methods
Author Response
Comment 1: This study proposes an improved YOLOv5 detection algorithm that can enhance the features in the image to make them more effective in mine and detect.
Response 1: Thank you for your comment.
Comment 2: Experimental results show that the proposed AC-YOLOv5 has achieved excellent results in accuracy, recall, and generalization of the model compared with the other 7 control groups experiment, especially the generalization for different datasets.
Response 2: Thank you for your comment.
Comment 3: Figure 1 is unclear.
Response 3: Thank you for pointing it out. We have replaced the original Figure 3 with a clear figure on page 3.
Comment 4: In Figure 3, coordinate axis units are ambiguous, and the specifications should be consistent for better comparison.
Response 4: Thank you very much for the significant comment. The Figure 3 is the training results from the YOLOv5 model owns result window. There are some items in the result window, including box (the position information of object, ie. center point, height, and width of box), obj (inside or outside the grid), cls (class information), metrics.
According to your suggestion, we have added relevant contents about the description of the result window. The relevant contents are highlighted on pages 6 & 7 of the manuscript in turquoise color.
Comment 5: Figure 3 needs to be detailed in the text to improve readability.
Response 5: Thank you very much for the significant comment. In order to state more clearly, according to your suggestion, we have added relevant contents in detailed explanation. The relevant contents are highlighted on page 6 of the manuscript in turquoise color.
Comment 6: It is recommended to elaborate in the Discussion what factors or differences make AC-YOLOv5 more accurate than the other 7 methods.
Response 6: Thank you very much for the valuable comment. We carefully studied the comment and added elaboration of factors about AC-YOLOv5 performance. The relevant contents are highlighted on page 8 of the manuscript in turquoise color, and updated relevant references.
